# Effects of Platelet-Rich Osteoconductive–Osteoinductive Allograft Compound on Tunnel Widening of ACL Reconstruction: A Randomized Blind Analysis Study

Ruth Solomon [1], Jan Pieter Hommen [2] and Francesco Travascio [1,3,4,5,*]

1. Department of Industrial and System Engineering, University of Miami, Coral Gables, FL 33146, USA; r.solomon5@umiami.edu
2. Department of Orthopedics, Florida International University, Miami, FL 33199, USA; jph@drhommen.com
3. Department of Mechanical and Aerospace Engineering, University of Miami, Coral Gables, FL 33146, USA
4. Department of Orthopaedic Surgery, University of Miami, Miami, FL 33136, USA
5. Max Biedermann Institute for Biomechanics at Mount Sinai Medical Center, Miami Beach, FL 33140, USA
* Correspondence: f.travascio@miami.edu

**Abstract:** The anterior cruciate ligament (ACL) is a commonly injured ligament in the knee. Bone tunnel widening is a known phenomenon after soft-tissue ACL reconstruction and etiology and the clinical relevance has not been fully elucidated. Osteoconductive compounds are biomaterials providing an appropriate scaffold for bone formation such as a demineralized bone matrix. Osteoinductive materials contain growth factors stimulating bone lineage cells and bone growth. A possible application of osteoinductive/osteoconductive (OIC) material is in ACL surgery. We hypothesized that OIC placed in ACL bone tunnels: (1) reduces tunnel widening, (2) improves graft maturation, and (3) reduces tunnel ganglion cyst formation. To test this hypothesis, this study evaluated the osteogenic effects of demineralized bone matrix (DBM) and platelet-rich plasma (PRP) on tunnel widening, graft maturation, and ganglion cyst formation. This was a randomized controlled clinical trial pilot study. A total of 26 patients that elected to have ACL reconstruction surgery were randomized between the OIC and control group. Measurements of tunnel expansion and graft-tunnel incorporation were conducted via the quantitative image analysis of MRI scans performed at six months after surgery for both groups. No patients had adverse post-operative reactions or infections. The use of OIC significantly reduced tunnel widening ($p < 0.05$) and improved graft maturation ($p < 0.05$). Patients treated with OIC had a significantly lower prevalence of ganglion cyst compared to the control group ($p < 0.05$). The use of OIC has measurable effects on the reduction of tunnel widening, improved graft maturation, and decreased size of ganglion cyst after ACL reconstruction. This study explored the utilization of biologics to minimize bone tunnel widening in ACL reconstruction surgery.

**Keywords:** demineralized bone matrix (DBM); MRI; bone healing; ganglion cyst; ACL graft maturation





## 1. Introduction

Anterior cruciate ligament (ACL) reconstruction surgery is a standard treatment with more than 100,000 procedures performed annually in the United States [1–6]. The surgery aims to restore knee stability and improve the patient's quality of life. Historically, the success rate of ACL surgery in returning patients to their previous level of sports activity ranges between 75–90% [7]. However, the process of rehabilitation after ACL injury can last for several months or even years, and represents a significant psychological and economic burden for the patient [8].

Bone tunnel widening is a known phenomenon after ACL reconstruction and has a significant correlation with the utilization of all-soft tissue grafts [9]. The largest percentage of tunnel widening occurs during the first six weeks after surgery and can continue over two years after surgery [10]. The incidence of tunnel widening ranges from 25–100% and

29–100% in the femoral and tibial tunnels, respectively [11–14]. Graft fixation implants including cortical fixation devices and bioabsorbable interference screws have been correlated with increased tunnel widening [15]. The etiology of widening is multifactorial, including mechanical factors (e.g., tunnel mal-positioning, graft motion wind-shield wiper effect, longitudinal elongation bungee-cording etc.), bone necrosis from the drilling technique, early rehabilitation, cytokine-induced bone resorption, and synovial fluid ingress into the bone tunnels [9,15]. Although the correlation of tunnel widening and clinical outcomes remains unclear, significant widening of the tunnels may complicate revision surgery [16].

Osteoinductive/osteoconductive compounds (OIC) are biomaterials characterized by bioactive properties: they provide an appropriate scaffold for bone formation (osteoconductivity) and are able to bind and concentrate endogenous bone morphogenetic proteins (BMP) (osteoinductivity) in circulation, thus promoting osteogenesis [17]. The use of demineralized bone matrix (DBM) has gained popularity in ACL reconstruction. DBM is a type I collagen matrix of allograft bone that remains after the extensive process of removing blood, cells, and minerals. The final result is a small particulate that can be applied to form a three-dimensional scaffold (osteoconductive) with osteogenic growth factors like BMPs, a family of transforming growth factor-β (TGF-β). Moreover, autologous platelet-rich plasma (PRP) is a concentrated solution of a patient's own platelets (or thrombocytes). By centrifuging the patient's whole blood, the red blood cells can be separated from the serum containing the concentrated platelets. Activated platelets have demonstrated osteoinductive properties on mesenchymal stem cells [18]. A possible application of DBM and PRP is to facilitate the osseointegration of the ACL graft into the bone tunnels to minimize tunnel widening and improve the structural stability of the graft construct. This in turn may allow the knee joint to adapt to bearing physiological mechanical loads sooner, ultimately with faster recovery of the patient.

As an initial step to validate our research hypothesis, this pilot study aimed to observe the osteogenic effects of DBM and PRP on tunnel widening, graft maturation, and tunnel ganglion cyst formation relative to control groups.

## 2. Materials and Methods

### 2.1. Patients Allocation

Twenty-seven patients (twenty-eight knees) were prospectively enrolled in a randomized controlled trial from 2016 to 2019. Patients were randomized to the OIC group after electing to have surgery. There were a total of 13 patients in the OIC group and 14 in the control group. There were 14 males and 12 females with an average age of 31 years old. One patient declined to participate in the study after undergoing treatment, and four others were lost to follow-up MRI. The Reporting Trials flow diagram is shown in Figure 1. The demographic data of all participants are summarized in Table 1. The experimental protocol was approved by the Institutional Review Board at the University of Miami. All subjects provided written consent prior to participation in the study. All the surgical procedures were performed by JPH. Data analysis was conducted at the University of Miami and the Hommen Orthopedic Institute.

**Table 1.** Summary of the demographic data of participants.

|  | OIC (StimuBlast®/PRP) | Control |
|---|---|---|
| **Male** | 8 | 7 |
| **Female** | 5 | 7 |
| **Age at Time of Surgery** | 32.7 ± 10.5 | 32.7 ± 12.7 y.o. |
| **Age Range** | 16 to 50 y.o. | 15 to 55 y.o. |

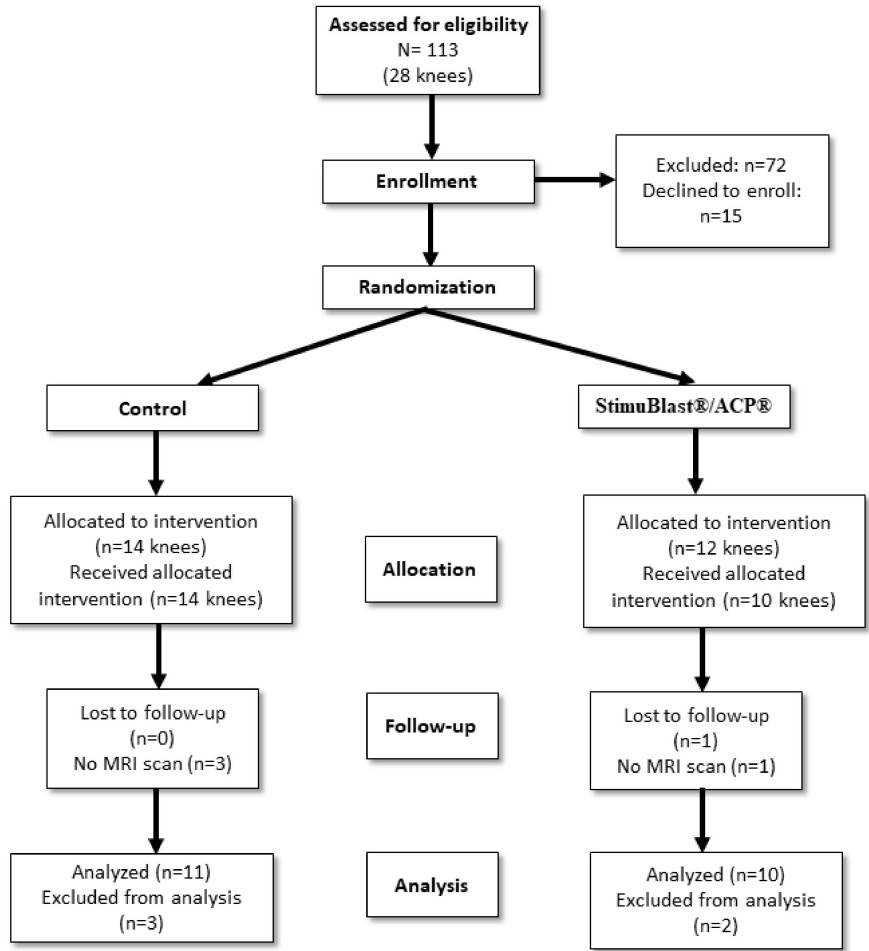

**Figure 1.** The chart shows the Consolidated Standards of Reporting Trials flow diagram for this randomized controlled trial with intention to treat analysis.

### 2.2. Inclusion and Exclusion Criteria

The inclusion criterion was indication for primary anterior cruciate ligament reconstruction with all-soft tissue allografts fashioned into a GraftLink® (Arthrex, Inc., Naples, FL, USA) construct with cortical button fixation on the tibia and femur. The exclusion criteria were revision surgery, multi-ligament surgery, autografts, and patients under 18 years of age. Excluded patients were: 32 autografts, 9 ACL repairs, 7 revisions, 5 multi-ligaments, and 19 with interference screw fixation. Patients with concomitant procedures were included: 14 with meniscus repair (9 OIC, 5 control), 4 with partial meniscectomy (1 OIC, 3 control), and 3 with chondroplasty (0 OIC, 3 control).

### 2.3. Surgical Procedures

Patients randomized to the OIC group had blood drawn in the pre-op holding area, which was subsequently prepared utilizing the Arthrex ACP® system to yield platelet-rich plasma (PRP). The PRP was mixed with 5 mL of StimuBlast® (Arthrex, Inc., Naples, FL, USA). The graft was constructed using an allograft peroneus longus tendon with an ACL Tightrope® button (Arthrex, Inc., Naples, FL, USA) at each end. The femoral and tibial tunnels were reamed to match the graft size. We utilized a low-profile reamer through a medial portal to create the femoral tunnel and a FlipCutter® (Arthrex, Inc., Naples, FL, USA) for the tibia tunnel. Prior to docking the graft into the femoral and tibial sockets in the OIC group, 1 to 2 mL of mixed PRP/DBM was injected via a syringe into each tunnel with the arthroscopic irrigation turned off to avoid extravasation. The graft was then secured into the sockets utilizing the tension-slide button mechanism. Additional

PRP/DBM was injected into both tunnels at the graft–bone interface after securing the graft. Concomitant meniscus pathology was addressed by meniscectomy or repaired utilizing the surgeon's choice of an all-inside, inside-out, or outside-in technique. Chondral injuries were addressed with chondroplasty. A hinged knee brace was placed post-operatively for three weeks for isolated ACL reconstructions and six weeks for concomitant meniscus repairs. Immediate weight bearing as tolerated versus partial foot flat weight bearing was allowed for isolated ACL reconstructions and meniscus repairs, respectively.

### 2.4. MRI Scanning Protocol

The intra-articular graft, tunnels, and graft–tunnel interface were assessed using a 1.5T MR System (Optima MR430s; GE Healthcare) with sequences in the sagittal and coronal planes, as well as axial cuts that are perpendicular to the femoral and tibial tunnels. A summary of the parameters used in the MRI scanning protocol is reported in Table 2.

**Table 2.** Summary of MRI scanning parameters by image type. Parameters shown are repetition time (TR), echo time (TE), inversion time (TI), echo train length (ETL), matrix size, field of view (FOV), and slice thickness (with 0 mm gaps).

| | TR/TE/TI (ms) | ETL | Matrix Size | FOV (mm) | Slice Thickness (mm) |
|---|---|---|---|---|---|
| **FSE Sagittal PD** | 3200/30 | 10 | 512 × 256 | 140 | 4 |
| **FSE Sagittal STIR** | 4000/36/130 | 10 | 256 × 192 | 160 | 4 |
| **FSE Sagittal (meniscus)** | 1500/19 | 10 | 256 × 224 | 140 | 3 |
| **FSE Coronal PD** | 3200/30 | 10 | 512 × 256 | 120 | 4 |
| **FSE Coronal STIR** | 4000/36/130 | 10 | 256 × 192 | 140 | 4 |
| **FSE Axial PD** | 3200/30 | 10 | 512 × 256 | 140 | 4 |
| **FSE Axial PD (Additional-perpendicular to graft tunnels)** | 3000/34–40 | 10 | 256 × 224 | 140 | 3 |

### 2.5. Measurement of Tunnel Widening

All measurements were conducted by a researcher blinded to treatment allocation prior to statistical analysis. Measurements of both tunnel expansion and graft maturation were conducted via quantitative image analysis of MRI scans performed at six months after surgery for both groups. Measurements of the tibial and femoral tunnel widening diameters were conducted using an established approach [19]. The femoral tunnel and tibial tunnels were measured at four points: the aperture, midpoint, tunnel end, and at the greatest tunnel diameter by taking the measurement of the diameter perpendicular to the tunnel—see Figure 2A–F. The tunnel expansion was calculated in the femur and tibia by subtraction of the initial surgical tunnel diameters from the largest measurement taken in each respective region. The quantity of patients with moderate (5 mm) and large (10 mm+) tunnel expansion was also noted for each group.

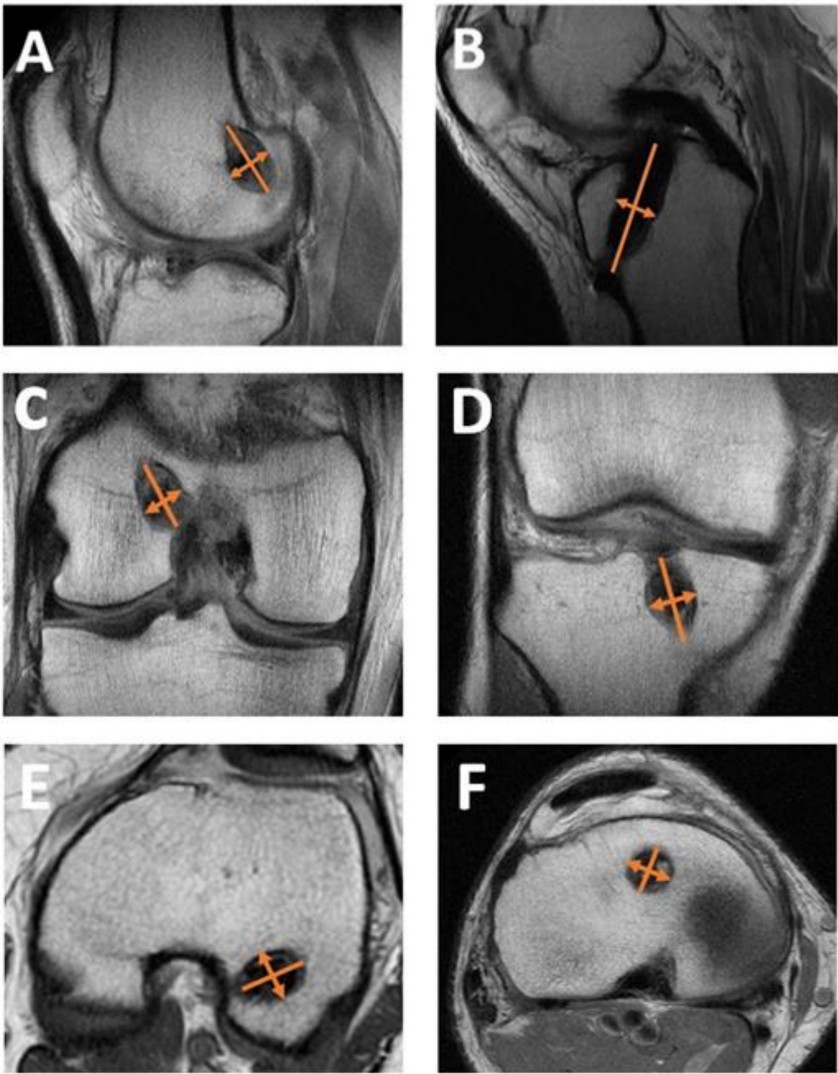

**Figure 2.** Measurements of tunnel expansion are shown in diagrams A-F: the femoral tunnel diameter was determined in each region of interest by taking measurements of the diameter perpendicular to the greatest diameter in the (**A**) sagittal, (**C**) coronal, and (**E**) axial MRI images; the tibial tunnel diameter was measured in each region of interest by taking measurements of the diameter perpendicular to the tunnel in the (**B**) sagittal, (**D**) coronal, and (**F**) axial MRI images.

*2.6. Measurement of Graft Maturation*

Graft maturation was quantified using a previously reported approach to evaluate the signal/noise quotient (SNQ) of MRI images by taking the average intensity across three areas of the ACL graft: the proximal, central, and distal intra-articular regions [20–22]—see Figure 3. The measurement of SNQ was carried out as follows: in correspondence with each region where the measurement was performed, a circular region of interest with a 5-mm diameter was defined and the signal intensity from the MRI scan was measured and averaged across the region ($S_{ROI}$). In a similar manner, the signal intensity ($S_{PCL}$) was also measured in the intact PCL with the purpose of normalizing the signal intensities measured in the ACL graft. To eliminate background noise, the signal intensity ($S_{BACK}$) was also measured at a background region of interest that was 2-cm anterior to the patellar tendon. The SNQ was then calculated using the following custom equation [23]:

$$SNQ = (S_{ROI} - S_{PCL})/S_{BACK} \tag{1}$$

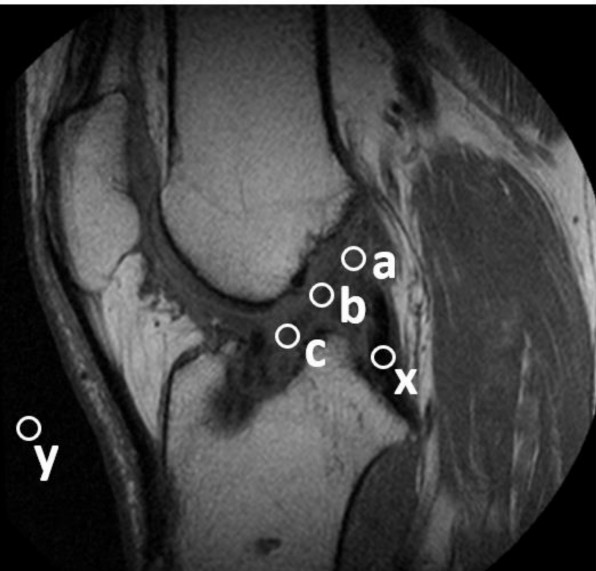

**Figure 3.** The locations of each region of interest used for the calculation of the SNQ are shown, including $S_{PCL}$ (x), $S_{BACK}$ (y), and the proximal (a), central (b), and distal (c) intra-articular regions, $S_{ROI}$.

### 2.7. Tunnel Ganglion Cyst Formation

Additionally, patients with tunnel ganglion cysts were quantified for each group by the blinded radiologist. Post-surgical graft failures were also assessed.

### 2.8. Statistical Analysis

All the statistical analyses were conducted in Minitab® 19.2 (Minitab. LLC, State College, PA, USA). A Mann–Whitney test was utilized to assess the differences in tunnel expansion associated with the OIC and control groups in both femur and tibia tunnels. A similar approach was used to infer possible differences in tunnel diameter measurements between the femur and tibia. A quantifiable difference in the proportion of ganglion cyst formation between the OIC and control groups was determined and compared via a Chi-Squared test with Yates correction. Differences in the measurements of SNQ were also investigated via Mann–Whitney test to compare OIC to control groups within the intra-articular region. For each test conducted, a level of significance of 0.05 ($\alpha = 0.05$) was used. All the data were reported in terms of mean ± standard deviation.

## 3. Results

No patients had adverse post-operative reactions or infections. One patient in the OIC group suffered graft failure 3 months after operation after falling on a boat. No other participants of the study presented with graft failure.

### 3.1. Tunnel Widening

A summary of the measurements of average tunnel expansion are reported in Figure 4. A summary of the tunnel expansion and their categorical sizes are shown in Tables 3 and 4. A significant difference was determined in femoral tunnel expansion ($p = 0.00736$) between the OIC ($1.00 \pm 0.64$ mm) vs. control group ($2.46 \pm 1.44$ mm). Additionally, a significant difference in tibial tunnel expansion ($p = 0.0001$) was found between the OIC ($1.08 \pm 0.63$ mm) and control group ($4.13 \pm 2.32$ mm). For the control group, the tibial tunnel expansion ($4.13 \pm 2.32$ mm) was found to be larger ($p = 0.046$) than the femoral tunnel expansion ($2.46 \pm 1.47$ mm). However, no significant difference was found between the femoral and tibial expansion of the OIC group ($p > 0.05$).

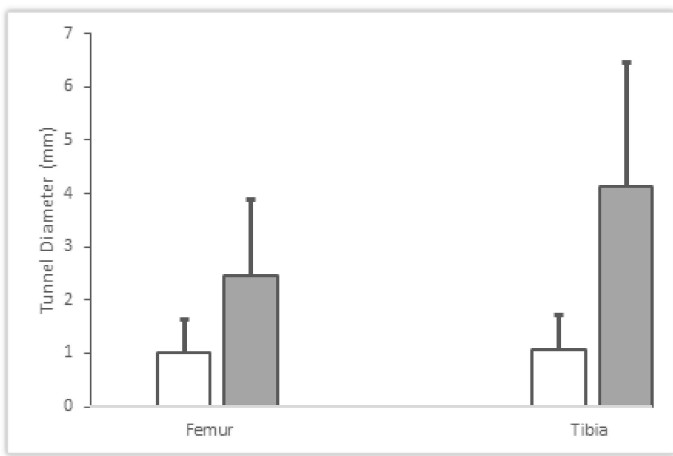

**Figure 4.** The figure shows the tunnel diameters measured from the post-operative MRI scans. Values for StimuBlast®/ACP® for the OIC (white) and control (grey) groups are reported for both the femur and tibia.

**Table 3.** Summary of femoral tunnel expansion by categorical size.

|  | OIC (StimuBlast®/ACP®) | Control |
|---|---|---|
| **Minimal Expansion (<1 mm)** | 5 | 0 |
| **Mild Expansion (<5 mm)** | 6 | 11 |
| **Moderate Expansion (5–10 mm)** | 0 | 1 |
| **Large Expansion (10+ mm)** | 0 | 0 |

**Table 4.** Summary of tibial tunnel expansion by categorical size.

|  | OIC (StimuBlast®ACP®) | Control |
|---|---|---|
| **Minimal Expansion (<1 mm)** | 4 | 0 |
| **Mild Expansion (<5 mm)** | 8 | 9 |
| **Moderate Expansion (5–10 mm)** | 0 | 2 |
| **Large Expansion (10+ mm)** | 0 | 1 |

*3.2. Graft Maturation*

A one-tailed Mann–Whitney test showed a decreased SNQ in the OIC group ($2.28 \pm 1.97$ mm) when compared to the control group ($4.28 \pm 2.55$ mm) ($p = 0.04363$).

*3.3. Tunnel Ganglion Cyst Formation*

Two of the ten OIC patients (20%) and eight of the eleven control patients (72.72%) presented with ganglion cysts in the tibial tunnel. The difference between these proportions was statistically significant ($p = 0.048$).

**4. Discussion**

Injury of the ACL may represent a significant psychological and economic burden to patients, given the lengthy recovery process [8]. There is significant research aimed at increasing the rate of graft-to-tunnel incorporation and graft maturation [24,25], as well as reducing the recovery time after surgery via innovative rehabilitation modalities [7]. The OIC materials have been heralded as osteogenic [17]. As further evidence, a recent clinical study with a follow-up at two years has shown that the use of DBM provides an internal brace to the ACL and represents a reliable and safe option when performing ACL construction [26]. Such a study was conducted on patients with a similar age range as of those participating in the present research. In addition, ACL fixations were conducted using an internal brace similar to that adopted in this study. This led us to hypothesize

that the injection of these materials into the bone tunnel would enhance graft incorporation and graft maturation. Aimed at testing our hypothesis, we compared the use of DBM combined with PRP in soft-tissue graft reconstructions to a control group and monitored the post-operative events of tunnel widening, graft maturation, and tunnel ganglion cyst formation through a quantitative analysis of MRI images. We adopted StimuBlast®, a DBM, due to its osteoconductive and -inductive properties. The DBM is manufactured with a reverse phase medium that allows it to be fluid-phased during handling and more viscous at body temperature. The DBM putty was augmented with anabolic properties of ACP®, a leukocyte-poor PRP, to promote the healing response during the phases of inflammation, cellular proliferation, and subsequent tissue remodeling. Unlike leukocyte-rich PRP, this does not induce potentially catabolic cytokines, such as interleukin-1β, tumor necrosis factor-α, and metalloproteinases, involved in the inhibition of bone-tunnel osteointegration [27]. The intra-operative application of DBM/PRP is relatively simple for the surgeon and assistants. The mixture is easily mixed and can be readily injected from a syringe through a curved needle applicator into the tunnels. Once placed, the mixture is viscous enough to remain in the tunnels and becomes more viscous at body temperature.

Tunnel widening has several implications after ACL surgery, including the possibility of post-operative tibia stress fractures [28–30] and delays or failures of graft incorporation [31]. In the setting of revision surgery, significant tunnel widening may require a bone grafting procedure followed by subsequently staged ACL revision surgery [10,16]. The presence of ganglion cysts within the tunnels has been associated with possible incomplete graft–bone incorporation [32,33]. Tunnel widening and ganglion cyst formation may represent a localized osteolytic process due to a cell-mediated cytokine response [34]. Our results indicate that the use of DBM and PRP: (1) significantly reduced tunnel widening, (2) had a positive effect on graft maturation, and (3) reduced the presence of ganglion cyst formation within the tibial tunnel.

It should be noted that the present contribution is a pilot study and its findings do not allow for causal relationships among the DBM/PRP injected and the post-operative outcomes to be established. Nevertheless, we postulate that the use of DBM/PRP may have enhanced the organization of the fibrocartilage insertion and mineralization, as well as reduced the cell-mediated osteolysis at the graft–bone interface. In animal studies, the use of DBM has been shown to create a 4-zone fibrocartilage tendon-to-bone healing [35]. In a rabbit model, Anderson et al. wrapped a BMP and TGF-β soaked collagen sponge around the graft inside the tunnel. Histologically, they observed more consistent bone-to-graft apposition and a fibrocartilaginous interface relative to controls at 2, 4, and 8 weeks. Biomechanically, the grafts also demonstrated significantly increased ultimate tensile strength at 2 and 8 weeks [1]. It has been proposed that synovial cytokines, such as TNF-α, interleukin 1β, IL-6, BMPs, and nitric oxide, may mediate ACL tunnel widening [34]. DBM/PRP may have served as a grout, limiting the ingress of synovial fluid into the interface between the graft and tunnel wall. The use of DBM/PRP may also have improved the location of graft fixation within the tunnel. In the control group, the graft relied solely on cortical suspensory fixation, which may have led to increased graft motion due to windshield wiper and bungee cord effects [36]. Instead, in the treatment group, the grafts were "potted" into a DBM/PRP putty that may have resulted in more aperture fixation and less micromotion. If DBM/PRP can enhance osteointegration of the graft within the tunnels, this may lead to earlier graft "ligamentization", or maturation, due to the increased revascularization of the implanted graft, as may have been demonstrated by the MRI signal intensity findings in our study [23,37]. It should be noted that a previous animal study on ACL reconstruction has shown that, while DBM alone promotes bone tunnel healing, the use of PRP alone does not produce any significant improvement with respect to a control group [38]. However, the above-mentioned study utilizes PRP concentrations and techniques of preparation different from those we adopted. This may explain the discrepancy of the results reported in the findings reported in Hexter et al. [38] with the present study.

Several studies have investigated the effects of injectable OICs in tendon-to-bone healing. Animal studies conducted on rats [39], dogs [40], rabbits [41,42], and goats [43] have found accelerated graft-to-bone healing rates with calcium phosphate. In fact, Ma et al. demonstrated that calcium phosphate could deliver increased local BMPs in tendon-to-bone healing [44]. Accordingly, in a randomized controlled human trial, Mutsuzaki et al. used calcium phosphate on 64 patients with follow-ups at 1 and 2 years [45]. At 1 year, tunnel expansion in patients treated with calcium phosphate was reduced by approximately 7% when compared to the control group. The use of calcium phosphate remains controversial due to its potentially slow degradation rate, which may slow osteointegration [46].

As a pilot study, we used a convenient sample of 26 patients, with the surgeries being performed by a single surgeon. A multicenter study, including a larger sample size and more surgeons, will be needed to confirm the preliminary evidence reported in this contribution. Another limitation of this study is the short observation period of less than a year, as a longer-term analysis may show larger differences in graft incorporation and maturation between the OIC and control groups. This issue will be addressed in a future work.

## 5. Conclusions

In conclusion, an in vivo study was conducted to evaluate the effects of an injectable OIC compound on the graft incorporation and tunnel widening after ACL reconstructive surgery. The results of this investigation indicate that the injection of OIC has measurable effects on the reduction of tunnel widening and ganglion formation and enhanced graft maturation after ACL reconstruction. Also, the benefits of augmenting ACL reconstruction with the injection of OIC may be observed within one year after surgery. Collectively, the use of osteogenic compounds may enhance ACL reconstruction graft incorporation and thereby promote earlier rehabilitation and return to sports.

**Author Contributions:** Conceptualization, J.P.H. and F.T.; methodology, F.T.; software, R.S.; formal analysis, R.S., J.P.H. and F.T.; investigation, J.P.H. and F.T.; resources, J.P.H. and F.T.; data curation, R.S.; writing—original draft preparation, R.S., J.P.H. and F.T.; writing—review and editing, R.S., J.P.H. and F.T.; supervision, J.P.H. and F.T.; project administration, F.T.; funding acquisition, J.P.H. and F.T. All authors have read and agreed to the published version of the manuscript.

**Funding:** The project described was supported by Grant Number GR004579 from Arthrex, Inc.

**Institutional Review Board Statement:** This study was registered at clinicaltrials.gov NCT04993339.

**Informed Consent Statement:** Informed consent was obtained from all subjects involved in the study.

**Data Availability Statement:** The data presented in this study are available on request from the corresponding author.

**Conflicts of Interest:** The authors declare no conflict of interest.

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
