# Peer review of "Effects of Platelet-Rich Osteoconductive–Osteoinductive Allograft Compound on Tunnel Widening of ACL Reconstruction: A Randomized Blind Analysis Study"

_pathophysiology, doi:10.3390/pathophysiology29030031_

Round 1
Reviewer 1 Report
The work described in the present manuscript presents the results of using a combination of demineralised bone matrix with platelet-rich plasma on ACL reconstruction. However, only one parameter was analised in the clinical trial, which makes the results section quite poor.
Another clinical trial using DBM for the same procedure shows a more complete set of results. This reference must be included and discussed in the present work: https://doi.org/10.1016/j.asmr.2021.07.030
Moreover, the use of combinations of DBM+PRP has been successfully demonstrated in animal models. The authors may want to include the reference: https://doi.org/10.1177/23259671211034166
Conclusions need to be presented in a dedicated section. Also, the authors claim that "A comparison of our findings to those of previous studies suggests that the benefits of augmenting ACL reconstruction with the injection of OIC may be observed within one year after surgery. " There are no results after one year to support this conclusion. At the very least, a reference with results from previous studies needs to be presented to validate this claim, and the claim must be clearly associated with the work of others, not as a conclusion of the present work.
As a final note, references are not numerically sequenced. This must be corrected.
Author Response
The work described in the present manuscript presents the results of using a combination of demineralized bone matrix with platelet-rich plasma on ACL reconstruction. However, only one parameter was analyzed in the clinical trial, which makes the results section quite poor.
Author’s response: We respectfully disagree with the Reviewer. Our study reports outcomes in terms of (1) quantitative assessment of tunnel widening, (2) graft maturation and (3) tunnel ganglion cyst formation. Similar studies (e.g. Lavender et al.2021) have reported clinical outcomes in terms of standardized score systems (e.g. Marx score, IKDC score, etc.). Such approach would be meaningful for observations carried out over a time period superior to 1 year. Unfortunately, the observation period of our study was 6 to 12 months, thus preventing us from including these additional metrics. While a follow-up over a longer period of time is in our future plans, we still believe that the information reported in this contribution is highly valuable, remarking the benefits of using a combination of DBM and PRP to enhance the clinical outcomes of ACL reconstruction.
Another clinical trial using DBM for the same procedure shows a more complete set of results. This reference must be included and discussed in the present work: https://doi.org/10.1016/j.asmr.2021.07.030.
Author’s response: We have included this new reference and discusses its findings in relation to our study. See lines 209-211.
Moreover, the use of combinations of DBM+PRP has been successfully demonstrated in animal models. The authors may want to include the reference: https://doi.org/10.1177/23259671211034166.
Author’s response: The reference as been added as suggested and the results reported in that contribution have been discussed considering the findings of our study. See lines 258-264
Conclusions need to be presented in a dedicated section.
Author’s response: Done as suggested: a Conclusions section has been added including the last portion of Discussion.
Also, the authors claim that "A comparison of our findings to those of previous studies suggests that the benefits of augmenting ACL reconstruction with the injection of OIC may be observed within one year after surgery. " There are no results after one year to support this conclusion. At the very least, a reference with results from previous studies needs to be presented to validate this claim, and the claim must be clearly associated with the work of others, not as a conclusion of the present work.
Author’s response: A reference has been added as suggested, see Line 287. The work referenced indicates positive clinical outcomes in bone tunnel healing upon use of DBM and bone marrow after two years a follow up.
As a final note, references are not numerically sequenced. This must be corrected.
Author’s response: Noted. We will work with the technical editor to fix this issue.
Reviewer 2 Report
The article entitled “Effects of Platelet Rich Osteoconductive-Osteoinductive Allograft Compound on Tunnel Widening of ACL Reconstruction: A Randomized Blind-Analysis Study” this pilot study aimed to observe the osteogenic effects of DBM and PRP on tunnel widening, graft maturation, and tunnel ganglion cyst formation relative to control groups.
This article deals with new alternatives for knee ligament problems, a fact of great clinical importance. Below are some suggestions:
In the Abstract:
The abstract is well written, but when the authors mention the demineralized bone hue, they could already mention which one was used in the research. The methodology could be better described.
In the Introduction:
On line 53, when talking about biomaterials, I suggest, as in the abstract, talking about the matrix, in the specific case of the research, the peroneus longus tendon.
In the Materials and Methods:
- The figure and also Table 1 have very clear and objective information.
- In line 157, was the equation used suggested by the program? Better explain the measures used.
In te Discussion:
- As it is a pilot study, the discussion is well outlined because it is difficult to compare with data from the literature. The limitations are described and the conclusion is in accordance with the hypothesis presented.
Author Response
The article entitled “Effects of Platelet Rich Osteoconductive-Osteoinductive Allograft Compound on Tunnel Widening of ACL Reconstruction: A Randomized Blind-Analysis Study” this pilot study aimed to observe the osteogenic effects of DBM and PRP on tunnel widening, graft maturation, and tunnel ganglion cyst formation relative to control groups.
This article deals with new alternatives for knee ligament problems, a fact of great clinical importance. Below are some suggestions:
In the Abstract: The abstract is well written, but when the authors mention the demineralized bone hue, they could already mention which one was used in the research. The methodology could be better described.
Author’s response: Since this study was partially funded by the company producing the DBM, we preferred to limit explicitly mentioning the commercial name of the biologic as much as possible. Also, due to space limitations, we deferred the details on the methodology adopted to the main text of the paper to digress more on the findings of the paper and its implications.
In the Introduction: On line 53, when talking about biomaterials, I suggest, as in the abstract, talking about the matrix, in the specific case of the research, the peroneus longus tendon.
Author’s response: As explained in the above comment, we limited, as much as possible, any elaboration on the specific compound use. The information reported in the paragraph starting on line 53 was introduced to illustrate the advantages of using a DBM without any specific bias towards any specific commercially available biologic.
In the Materials and Methods: - The figure and also Table 1 have very clear and objective information. In line 157, was the equation used suggested by the program? Better explain the measures used.
Author’s response: In order to improve the clarity of the information provided, on line147, we have specified that the analysis is conducted on MRI images. We have also specified that the equation used for calculating the SNQ metric was custom developed according to previous studies, see line 156.
In the Discussion: - As it is a pilot study, the discussion is well outlined because it is difficult to compare with data from the literature. The limitations are described and the conclusion is in accordance with the hypothesis presented.
Author’s response: We thank the reviewer for appreciating our effort. Following the suggestion of Reviewer #1, we have enriched our discussion with comparison with the most recent study (Lavender et al. 2021, Hexter et al. 2021) reporting positive outcomes when reconstructing ACL augmented with DBM, see Lines 208-2011 and 258-264.
Round 2
Reviewer 1 Report
The work described in the present manuscript presents the results of using a combination of demineralized bone matrix with platelet-rich plasma on ACL reconstruction. However, only one parameter was analyzed in the clinical trial, which makes the results section quite poor.
Author’s response: We respectfully disagree with the Reviewer. Our study reports outcomes in terms of (1) quantitative assessment of tunnel widening, (2) graft maturation and (3) tunnel ganglion cyst formation. Similar studies (e.g. Lavender et al.2021) have reported clinical outcomes in terms of standardized score systems (e.g. Marx score, IKDC score, etc.). Such approach would be meaningful for observations carried out over a time period superior to 1 year. Unfortunately, the observation period of our study was 6 to 12 months, thus preventing us from including these additional metrics. While a follow-up over a longer period of time is in our future plans, we still believe that the information reported in this contribution is highly valuable, remarking the benefits of using a combination of DBM and PRP to enhance the clinical outcomes of ACL reconstruction.
Reviewer’s comment (round 2): A bit more detailed presentation of results on points 2 and 3 was expected, but it is admissible providing the topics are adequately detailed in the discussion. Regarding observation period, it is important to mention the plans of follow-up studies in the conclusions section as a perspective for future work.
Another clinical trial using DBM for the same procedure shows a more complete set of results. This reference must be included and discussed in the present work: https://doi.org/10.1016/j.asmr.2021.07.030.
Author’s response: We have included this new reference and discusses its findings in relation to our study. See lines 209-211.
Reviewer’s comment (round 2): As done with ref [45], a comparison of the tested conditions for this reference would help confirm that the results are comparable and further validate the conclusions. One of the questions raised by this publisher (MDPI) during the reviewer’s examination of the manuscript is whether the results support the conclusions. It is difficult to reply “100% yes” when the conclusions mention long-term outcomes that are missing from the present study and only available in related studies from the literature.
Moreover, the use of combinations of DBM+PRP has been successfully demonstrated in animal models. The authors may want to include the reference: https://doi.org/10.1177/23259671211034166.
Author’s response: The reference as been added as suggested and the results reported in that contribution have been discussed considering the findings of our study. See lines 258-264
OK, perfect!
Conclusions need to be presented in a dedicated section.
Author’s response: Done as suggested: a Conclusions section has been added including the last portion of Discussion.
Also, the authors claim that "A comparison of our findings to those of previous studies suggests that the benefits of augmenting ACL reconstruction with the injection of OIC may be observed within one year after surgery. " There are no results after one year to support this conclusion. At the very least, a reference with results from previous studies needs to be presented to validate this claim, and the claim must be clearly associated with the work of others, not as a conclusion of the present work.
Author’s response: A reference has been added as suggested, see Line 287. The work referenced indicates positive clinical outcomes in bone tunnel healing upon use of DBM and bone marrow after two years a follow up.
Reviewer’s comment (round 2): Again, conclusions should be based on the results from the present work. Adding to that, literature results can help build a perspective of outcome within a broader research overview, but they cannot be used to build the conclusions of the present work. These two concepts (conclusions drawn from own results vs literature results) must be clearly separated in section 5.
As a final note, references are not numerically sequenced. This must be corrected.
Author’s response: Noted. We will work with the technical editor to fix this issue.
OK
Author Response
Responses to Reviewer’s Comments
- Reviewer’s comment (round 2): A bit more detailed presentation of results on points 2 and 3 was expected, but it is admissible providing the topics are adequately detailed in the discussion. Regarding observation period, it is important to mention the plans of follow-up studies in the conclusions section as a perspective for future work.
Authors’ Response: Done as suggested, see Line 280.
- Reviewer’s comment (round 2): As done with ref [45], a comparison of the tested conditions for this reference would help confirm that the results are comparable and further validate the conclusions. One of the questions raised by this publisher (MDPI) during the reviewer’s examination of the manuscript is whether the results support the conclusions. It is difficult to reply “100% yes” when the conclusions mention long-term outcomes that are missing from the present study and only available in related studies from the literature.
Authors’ Response: Similarities in the methods used in study [46] and the present contribution have been noted, see Lines 210-212. We specified that both studies were conducted on patients of similar age range, and adopted a similar internal brace for the fixation of the ACL.
- Reviewer’s comment (round 2): Again, conclusions should be based on the results from the present work. Adding to that, literature results can help build a perspective of outcome within a broader research overview, but they cannot be used to build the conclusions of the present work. These two concepts (conclusions drawn from own results vs literature results) must be clearly separated in section 5.
Authors’ Response: The sentence in the conclusion section has been modified to solely reflect the summary of the findings reported in this study. Any reference to other works in the literature has been deleted. The new sentence reads: “the benefits of augmenting ACL reconstruction with the injection of OIC may be observed within one year after surgery”, see lines 288-289.